# Effectiveness of Cervical Testing in and outside a Screening Program—A Case-Control Study

**DOI:** 10.3390/cancers14215193

**Published:** 2022-10-23

**Authors:** Maiju Pankakoski, Tytti Sarkeala, Ahti Anttila, Sirpa Heinävaara

**Affiliations:** 1Finnish Cancer Registry, Unioninkatu 22, 00130 Helsinki, Finland; 2Faculty of Medicine, University of Helsinki, Haartmaninkatu 8, 00014 Helsinki, Finland

**Keywords:** cervical cancer, screening, effectiveness study, case-control study

## Abstract

**Simple Summary:**

Opportunistic cervical testing is a common practice in many countries. However, quality control of tests outside national screening programs is often not possible. We conducted a case-control study on the effect of cervical testing in and outside the Finnish screening program on the risk of cervical cancer. Women undergoing cervical tests in five- or three-year intervals had a lower risk of cancer compared with those not tested. Testing was effective in and outside the screening program. However, program tests should be preferred for cost-effectiveness. Our results also strengthened the previous findings in the literature, that cervical screening is most effective in women aged 35 and over, and preventive effects can be seen until very old ages.

**Abstract:**

In many countries with organized cervical cancer screening, opportunistic Pap and human papillomavirus (HPV) tests are common. However, little is known about their effectiveness. We examined the effect of testing in and outside the Finnish screening program on the risk of cervical cancer. We conducted a case-control study that involved 1677 cases with invasive cervical cancer that were diagnosed between 2010 and 2019. Five- and three-year test intervals were analyzed across all ages, by age group and by cancer morphology subtype. Conditional logistic regression was used, adjusting for socioeconomic variables. Women undergoing any kind of cervical test had a significantly lowered risk of cervical cancer (adjusted OR = 0.43, 95% CI = 0.38–0.48, tests in five-year intervals). The results were similar, regardless of whether the test had been taken in the screening program or outside of it, or whether the interval was five years or three years. Testing of women at ages 35–64 showed the strongest effects, but moderate preventive effects were seen until age 79. No significant effect was seen below age 30. Tests in and outside the program were effective at the screening target age. However, participation in the program should be encouraged for optimal cost-effectiveness. Preventive effects were also seen above the program target ages.

## 1. Introduction

High-quality screening effectively reduces the incidence of invasive cervical cancer [1,2,3]. This is possible due to the natural history of the disease: it develops over the years through a precancerous stage, preceded by a persistent infection with high-risk human papillomavirus (HPV) [4,5]. The precancer can be detected by screening and treated before it possibly progresses to cancer. The introduction of HPV vaccination has revolutionized cervical cancer prevention [6,7,8,9]. However, screening remains an important secondary prevention strategy, especially in older unvaccinated cohorts. 

In many countries with well-established population-based cervical cancer screening programs, Pap and HPV testing outside the programs is also common [10]. Proper quality control and management of such tests is often not possible.

Tests outside a screening program, or opportunistic tests, are generally common among younger women [11,12,13,14]. However, women below age 25 hardly benefit from tests in or outside the program [15,16,17], whereas older women continue to benefit from screening even above the usual screening upper-age limit of 60 to 65 [17,18,19]. 

Women undergoing cervical testing are more likely to have higher education levels, incomes, and socioeconomic status, as well as non-immigrant backgrounds [20,21,22]. This seems to be most evident in countries without population-based screening programs [23,24]. Thus, women with good access to health services tend to have better test coverage than less-privileged women.

Earlier studies suggested that there is no benefit from additional cervical testing for women tested in a screening program [25,26]. However, until now, no large-scale studies of tests in and outside a program have been conducted, due to a lack of comprehensive registered data.

We performed a register-based case-control study of the preventive effect of cervical testing on the risk of cervical cancer in the general population. We obtained 10 to 19 years of test data on each subject, both in and outside the Finnish screening program. We examined overall and age-specific effects of five- and three-year test intervals. The main results were reported by cancer morphology subtype, and socioeconomic variables were used to account for self-selection.

## 2. Materials and Methods

We identified 1677 cases from the Finnish Cancer Registry. These cases were women who were born between 1917 to 1994 and diagnosed with invasive cervical cancer (ICD-O-3 topography codes C53) [27] from 2010 to 2019 while aged between 21 years and 97 years. In each case, 10 controls from Finland’s Population Information System were individually matched by year of birth and residential area (hospital district) at the time of the case’s diagnosis (index date). Controls were sampled from those subjects who, at the index date, were alive, free of invasive cervical cancer, and residing in Finland, and who had not had their cervices removed. 

Data on tests in the screening program between 2000 and 2019 were derived from Finland’s Mass Screening Registry. Data on tests outside the screening program were available from 2000 until 2014 and to a limited extent until 2016, and were derived from pathology laboratories, Finland’s Health Insurance Reimbursement Register (Kela), and student health services (YTHS). 

Data on socioeconomic status and education at the index date were derived from Statistics Finland, and data on mother tongues came from the Population Information System. Total or partial hysterectomy records from 1990 to 2019 were derived from Finland’s Care Register for Health Care (HILMO).

During the study period, the Finnish screening program personally invited women aged 30 to 60 for a cervical test every five years. In some municipalities, women aged from 25 to 30 and/or 60 to 65 were also invited. Pap testing was the primary screening method in most parts of the country; approximately 7% of the program tests were HPV tests. In cases of mild abnormalities (ASC-US/HPV+), a follow-up invitation was sent after 1 to 2 years. More severe screening results led to colposcopy examinations. Information on the classification of different Pap results can be found, for example, in the Bethesda System for Reporting Cervical Cytology [28].

We analyzed the association between testing in a five- or three-year interval and cervical cancer diagnosis in the following five-year interval, using conditional logistic regression [29]. The main areas of interest were cervical testing by any test (in or outside the screening program), or by different modes of testing. The mode of testing was classified in four categories: only tests in the program; only tests outside the program; tests both in and outside the program; and no tests.

To evaluate the preventive effects of testing, we excluded from the analyses tests that had been taken less than 12 months before a case’s diagnosis (index date). Therefore, we disregarded any tests that could be part of an episode that led to cancer diagnosis.

The analysis of the five-year intervals, or the testing age bands, was straightforward, due to our five-yearly screening program. For example, we examined cancers in women at ages 35 to 39 against testing of women at ages 30 to 34. Some data exclusions were needed for the three-year age bands to ensure the inclusion of the program invitations. These age bands were chosen to start at ages that were divisible by five, as in the five-year analysis. As a result, some test ages were ignored. For example, we examined cancers in women at ages 33 to 37 and 38 to 42 against testing at ages 30 to 32 and 35 to 37. The definition of testing and the index age bands are described in Figure 1.

Descriptive statistics on the number of tests (one test vs. two or more) by mode of testing in the five-year testing age bands were reported for women at ages 15 to 89. The numbers were shown separately for controls and cases.

The overall effect of having any type of a test was analyzed for all ages. The effects by mode of testing were analyzed for all women at the screening target age (30 to 64 years). These analyses were further performed by cancer subtype by ICD-O-3 morphology, separately on squamous cell carcinoma (including microinvasive), fully invasive squamous cell carcinoma, and adenocarcinoma. Finally, the effects were reported separately by the five- and the three-year age bands for ages 20 to 84. No adjustments for multiple comparisons were performed.

To account for potential self-selection, models were adjusted for education, socioeconomic status, and mother tongue. Education was classified as tertiary, upper secondary, and basic or unknown. Socioeconomic status was classified as employees (including upper/lower-level employees, employers, and self-employed), manual workers, and others (including students, pensioners, others, and unknown). Mother tongue was classified as native (Finnish, Swedish, or Sami) and others. Only education was used as an adjusting variable for women screened at ages 65 or older, because the older age groups consisted almost entirely of pensioners and native speakers.

Sensitivity analysis was conducted for restricted data from 2010 to 2015. This was done to determine the possible bias caused by the incomplete registration of tests in the latest years of the follow-up.

## 3. Results

Sample characteristics are presented in Table 1. Cases were more likely to have basic or unknown education levels than controls. In addition, manual workers, women with unknown/other socioeconomic status, and women whose mother tongue was other than native were over-represented in the cases’ group.

The number of tests by mode of testing in the five-year testing age bands are presented in Figure 2. Tests outside the program and having more than one test during the five/three-year age band were most popular among the youngest age groups for both cases and controls. Test coverage and intensity among the cases started to decline above age 35, especially, and were thereafter notably lower than among controls. Those tested only outside the program were more likely to have two or more tests during the five-year age band. Overall, a significant proportion of cases had tests both in and outside the program.

The overall five-year test coverage preceding the index age band, pooled across all ages, was 54% for cases and 71% for controls. Coverage at the national screening target ages (30 to 64) was 62% for cases and 83% for controls (Table 2). A cervical test by any mode of testing resulted in a 57% reduction in the risk of cervical cancer (adjusted OR = 0.43, 95% CI = 0.38–0.48, Table 2). All modes of testing reduced the risk significantly, with overlapping confidence intervals. Having only program tests resulted in a 65% reduction in the cervical cancer risk (adjusted OR = 0.35, 95% CI = 0.29–0.41). Having tests only outside the program or having both types of tests resulted in a 58% and a 69% decrease in the risk (adjusted OR = 0.42, 95% CI = 0.35–0.51 and 0.31, 95% CI = 0.25–0.37), respectively. Any mode of cervical testing in the three-year interval resulted in a quite similar risk reduction as that resulting from testing in the five-year interval (adjusted OR = 0.49, 95% CI = 0.44–0.55, Table 2). Again, all modes of testing reduced the risk significantly.

Squamous cell carcinoma was the most common type of cancer (fully invasive 53%, microinvasive 6%), followed by adenocarcinoma (31%), other or unspecified carcinoma (9%), and sarcoma or unspecified malignancy (1%). The effects of testing in a five- and three-year interval by morphology are presented in Figure 3. Screening prevented squamous cell carcinoma better than it prevented adenocarcinoma. Excluding microinvasive tumors from the squamous cell cancer types resulted in a slightly better screening effect, although the difference was hardly noticeable. Results were very similar between the five- and three-year testing intervals (Figure 3).

Age-specific analyses showed no significant effect of testing by any mode on the adjusted risk of cervical cancer for women tested below age 30 (Figure 4). The results concerning testing during the early 30s were somewhat unclear. All modes of testing were most effective for women at ages 35 to 62/64. After the upper age limit of the screening program at 60 years (65 years in some areas), testing in the five- or three-year age bands reduced the risk of cervical cancer significantly up until age 79 (Figure 4). However, the proportion of tested women was small in the very oldest age bands.

Sensitivity analyses with a restricted follow-up time did not identify considerably differing results, although the odds ratios were somewhat smaller (Appendix A).

## 4. Discussion

Cervical testing in and outside the screening program prevented cancer effectively. Differences between the modes of testing were small and statistically insignificant. The results of the five- and three-year test intervals were similar, although odds ratios were slightly smaller with the five-year interval. Testing prevented all cancer types, but the effects were weaker against adenocarcinoma compared with the effects against squamous cell carcinoma.

Despite the frequent testing of young women, tests on women below the age of 30, by any mode, did not show a significant reduction in the risk of cancer. However, the number of cancer cases, especially in women below the age of 30, was small, and no program tests were performed for women below the age of 25. The preventive effect of testing was strongest for women who were aged approximately 35 to 64 years, and even tests for women above those ages prevented cancer up until old ages.

This was the first time that the effect of cervical testing on cancer risk was examined with comprehensive register data on all cervical tests, including those outside the screening program. We obtained individual-level data from various registries, with excellent coverage. We were able to exclude subjects who had had a hysterectomy and to control the analysis for self-selection bias. Screening histories for cases and controls were available for up to approximately 20 years.

However, some shortages concerning the tests outside the program existed, mostly for testing during the years 2014 to 2019. To account for this, we conducted sensitivity analyses with a restricted dataset, and we did not see major differences in the results. In addition, we were not able to define whether the outside tests were taken strictly for screening purposes or, instead, were based on symptoms. The exclusion of tests taken less than 12 months prior to the diagnosis corrected this problem in part. Still, symptom-based tests might have been taken even earlier, causing some selection among those women who were tested outside the program, possibly resulting in slightly weaker test effects. 

Data on hysterectomies were available beginning in the early 1990s. Therefore, some of this data on the very oldest cohorts might be inadequate, potentially leading to a slight overestimation of screening effects. The possible data shortages would apply to those born before the 1940s (approximately 15% of the study subjects). A peak in the prevalence of hysterectomies was earlier demonstrated for women born from 1943 to 1947 [30], indicating that most hysterectomies of this cohort were included in our data.

We controlled for selection-bias by adjusting the models for socioeconomic status, education, and mother tongue. In the age-specific analysis, we were able to control for education only in the oldest age bands. Because women tested above the target age were more likely to have high social status and no previous abnormal findings [31], it is likely that we were not able to fully control for selection bias at the oldest ages. 

The age-specific results were also affected by the invitational policy of the screening program. Women aged 25 and 65 years were only invited for screening in some parts of the country—for example, in the capital city, where the background risk for cervical cancer is higher compared to the risk in the rest of the country [19]. Matching the data by hospital district controlled for the differences between areas, but there could still be some discrepancies in the age-specific results for those age bands. In addition, the numbers of program tests for 25-year-old women and 65-year-old women were relatively small, resulting in large confidence intervals.

Our analysis was not directly comparable with most screening effectiveness studies, where exposure is defined as having a test within a certain interval prior to diagnosis [25,32,33]. First, the interval between the test and the index date varied, depending on when the diagnosis occurred in the following interval, or age band. Therefore, the period for the analysis of the five-year age bands could be anything from one to nine years. However, we assumed that the interval from test to diagnosis would, on average, be the length of the age band. Second, tests in the program were probably taken further from the index date, compared with tests outside the program, as the age band started with a screening invitation. Therefore, the time between the test and the diagnosis was somewhat longer for the program tests, potentially leading to weaker effects. In any event, this method allowed us to perform the age-specific analysis by focusing on the age at testing rather than the age at diagnosis [34]. For comparison, we analyzed the overall effect of testing using a different definition for screening exposure, considering tests that occurred within 5.5 or 3.5 years prior to the diagnosis. The results were quite well in-line with our main results, as presented in Appendix A.

Few studies on the extent and relevance of tests outside screening programs, i.e., opportunistic tests, exist in the literature. Tranberg et al. argued that opportunistic testing could cover non-participating women, but more frequent testing than recommended did not show additional benefits [13]. In our previous study, some ethnic minorities seemed to be better included in tests outside the program, although the overall coverage in these groups was low [11]. Special attention should be paid to these minorities to ensure sufficient test coverage—whether in or outside the program.

The screening interval in Finland is five years, which is wider than the screening interval in many countries. With the emergence of HPV testing, other countries are extending or considering extending screening intervals to five years, as the risk after a negative HPV test within five years is very small [15]. According to our results, mostly based on Pap tests, the five-year screening interval had at least as good a preventive effect as the three-year interval. On the other hand, when the screening interval is as wide as five years, women tend to be tested opportunistically between screens [14]; this applies to women in Finland.

Our analysis showed smaller odds ratios for squamous cell carcinoma than for adenocarcinoma, a result that was also demonstrated in previous literature [35,36]. Research suggests that HPV testing provides greater protection against adenocarcinoma, compared with the protection against cytology [37,38]. In our study, we did not compare Pap testing with HPV testing, as we did not have information on the primary test methods outside the program. In addition, as mentioned, HPV testing was relatively uncommon during the period of this study. However, at the time of writing this paper, HPV-based screening had become predominant in our country. The transition to HPV screening is justified because of its higher sensitivity for high-grade cervical lesions [39]. At the same time, HPV screening lacks specificity. Carefully designed triage methods are needed to avoid unnecessary colposcopy referrals and the detection of non-progressive lesions [40,41]. The impact of HPV screening on cancer incidence and mortality, as well as protection against adenocarcinoma, remains to be seen in future effectiveness studies.

Our age-specific results were quite well in line with previous studies, with no detectable effect of screening for women aged 25 or younger [16,17]. However, conflicting results have been reported [32,33]. Some studies suggested better effectiveness against more advanced cancers in younger women [42]. Nevertheless, an increasing incidence of cervical cancer among younger women has been demonstrated in developed countries in recent years [43]. In the future, HPV vaccination is likely to be the most efficient means of reducing the cancer burden among young cohorts [6,7]. In Finland, the HPV vaccination program was introduced for girls in 2013, and for boys in 2020.

Screening women at age 65 or older has shown promising results in previous studies [19,44]. However, low cancer risk has been demonstrated in women who have had only negative results after the age of 50 [18,45], indicating that risk stratification among older women could be considered. Our results showed the preventive effects of cervical testing for women at age 65 and older, up until the testing age band of 75 to 79. However, more research is needed on the effect of previous test history on cancer risk at the oldest ages.

Tests outside the screening program reduced cancer risk effectively, together with the program tests at the screening target ages. This indicated that the tests have good quality and lead to further examinations and treatments if necessary. The cancer burden in the population is most likely reduced by the screening program, together with other tests, as well as by improved cancer treatments. On the other hand, it should be noted that this study took place in a high-resource setting, with clinical practices of a good general quality. The quality of test practices, especially opportunistic testing, may vary greatly across Europe and the world. 

In addition, the cost-effectiveness of outside tests may be poorer than the cost-effectiveness of screening program tests, because they are more expensive and are often assigned too frequently [12]. Outside tests are common among young women with low cancer risk and among those who also participate in the screening program [11]. Frequent testing outside the program does not improve effectiveness [13] and is more likely to result in false-positive results, causing anxiety and distress [46]. Therefore, the coverage of the program should be increased among all target-aged women, and tests outside the program should be conducted only when necessary, e.g., if a woman has not participated in the program or has relevant symptoms. In addition, those women above the screening target ages who have had an inadequate screening history or previous abnormal findings should remain under surveillance outside the program [15]. Finally, it is imperative that all tests, including those outside the screening program, be registered centrally, so that their performance and effectiveness can be continuously monitored.

## 5. Conclusions

The current screening policy in Finland, with an invitation to the screening program every five years, works well in preventing cervical cancer. Our results showed that testing both in and outside the program was effective. However, to achieve better cost-effectiveness, participation in the program should be encouraged, and additional testing should be targeted to carefully defined high-risk groups. It might be worthwhile to further extend the upper age limit for screening, but more research is needed on the risk of cancer associated with previous screening histories of the elderly.

## Figures and Tables

**Figure 1 cancers-14-05193-f001:**
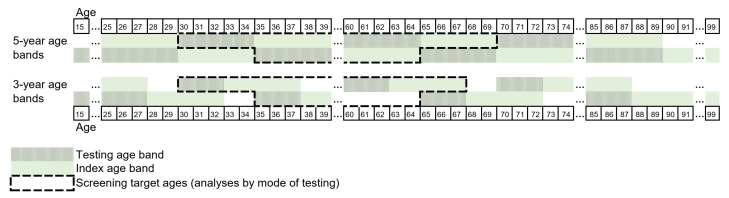
Definition of testing and index age bands. Testing in five- and three-year age bands against cervical cancers in the following five-year interval were examined. For example, for the five-year testing age bands, association between testing of women at ages 30 to 34 and cancers in women at ages 35 to 39 were examined. Note: those tests that occurred in the index age band (35 to 39) before the possible diagnosis were not considered. In the mode-specific analysis, only tests taken at the target ages of the national screening program (ages 30 to 64) were included: i.e., index ages of 35 to 69 and 33 to 67.

**Figure 2 cancers-14-05193-f002:**
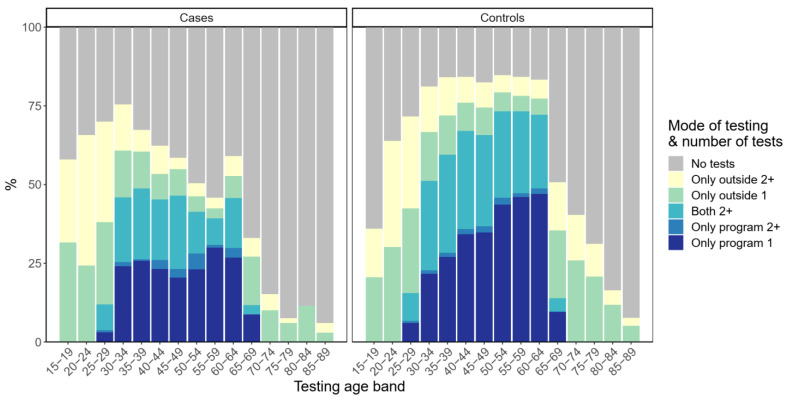
Test coverage and intensity by five-year testing age bands. Cases’ diagnoses occurred in the following five-year age band.

**Figure 3 cancers-14-05193-f003:**
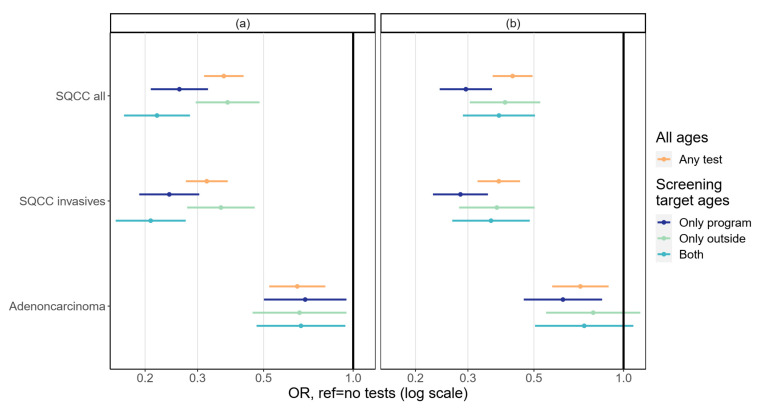
Association between testing in (**a**) a five-year interval and (**b**) a three-year interval and cervical cancer diagnosis at the following five-year interval by morphology. Adjusted for education, socioeconomic status, and mother tongue. SQCC = squamous cell carcinoma.

**Figure 4 cancers-14-05193-f004:**
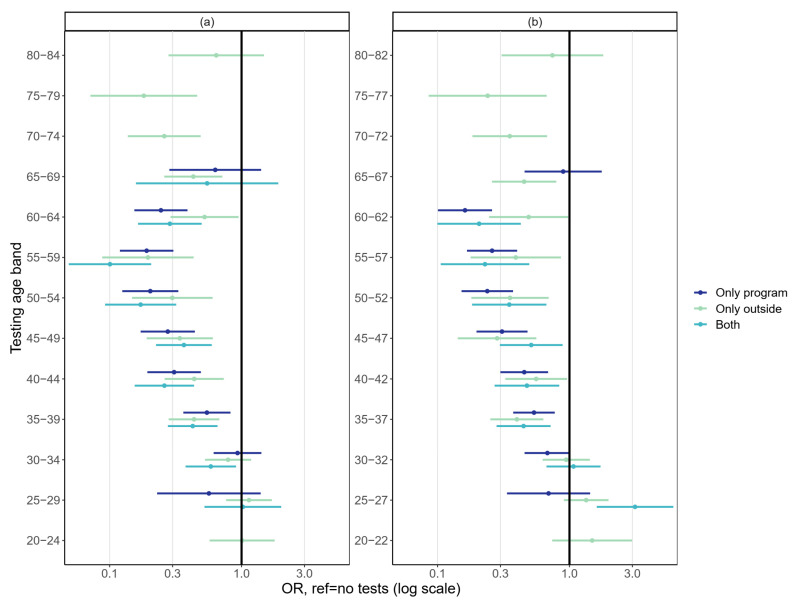
Association between testing by (**a**) a five-year testing age band and (**b**) a three-year testing age band and cervical cancer diagnosis in the following five-year interval. Adjusted for education, socioeconomic status, and mother tongue.

**Table 1 cancers-14-05193-t001:** Characteristics of the study sample.

Variable	Class	Cases, N (%)	Controls, N (%)
		1677 (100)	16,738 (100)
Age at case’s diagnosis	20–29	89 (5)	886 (5)
	30–39	380 (23)	3789 (23)
	40–49	351 (21)	3503 (21)
	50–59	263 (16)	2628 (16)
	60–69	247 (15)	2469 (15)
	70–79	182 (11)	1814 (11)
	80–89	127 (8)	1269 (8)
	90–99	38 (2)	380 (2)
Education	Tertiary	501 (30)	6516 (39)
	Secondary	645 (38)	6172 (37)
	Basic or unknown	531 (32)	4050 (24)
Socioeconomic status	Upper/lower-level employees	603 (36)	6962 (42)
	Employers and self-employed	73 (4)	737 (4)
	Manual workers	194 (12)	1615 (10)
	Students	44 (3)	374 (2)
	Pensioners	563 (34)	5568 (33)
	Others	200 (12)	1482 (9)

**Table 2 cancers-14-05193-t002:** Association between testing in a five- and three-year interval and cervical cancer diagnosis in the following five-year interval. Adjusted for education, socioeconomic status, and mother tongue.

	5-Year Interval			3-Year Interval		
	Cases, N (%)	Controls, N (%)	OR (95%CI)	Cases, N (%)	Controls, N (%)	OR (95%CI)
All ages	1677 (100)	16,738 (100)		1677 (100)	16,738 (100)	
No tests	779 (46)	4920 (29)		854 (51)	6063 (36)	
Any test	898 (54)	11,818 (71)	0.43 (0.38–0.48)	823 (49)	10,675 (64)	0.49 (0.44–0.55)
Screening target ages *	1081 (100)	10,793 (100)		1082 (100)	10,802 (100)	
No tests	412 (38)	1807 (17)		435 (40)	2178 (20)	
Only program	290 (27)	3864 (36)	0.35 (0.29–0.41)	362 (33)	5248 (49)	0.36 (0.31–0.43)
Only outside	181 (17)	2059 (19)	0.42 (0.35–0.51)	146 (13)	1685 (16)	0.48 (0.39–0.59)
Both	198 (18)	3063 (28)	0.31 (0.25–0.37)	139 (13)	1691 (16)	0.46 (0.37–0.56)

* The national screening target ages (in the whole country) at the time of study were 30, 35, 40, 45, 50, 55, and 60.

## Data Availability

Restrictions apply to the availability of these data. Data are available from the authors with the permission of the Finnish Institute for Health and Welfare.

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
