# Peer review of "Effectiveness of Cervical Testing in and outside a Screening Program—A Case-Control Study"

_cancers, 2022, doi:10.3390/cancers14215193_

Round 1
Reviewer 1 Report
In this manuscript, the authors examined the effect of testing in and outside the Finnish screening program on the risk of cervical cancer. They conducted a case-control study with 1677 cases with invasive cervical cancer diagnosed between 2010 and 2019. They performed a logistic regression by using five and three-year test intervals separately by age group and by cancer morphology subtype. They conclude that women undergoing cervical tests in five or three-year intervals had a lower risk of cancer compared with those not tested and testing was effective in and outside the screening program.
The study suffers major flaws in its design and conclusion. Cancer screening tests are shown to detect cancer early and reduce the chance of dying from that cancer. It is hard to understand that cancer screening is associated with a lower risk of cancer.
The manuscript is not of sufficient novelty to be published. The manuscript does not make any significant improvement to the intended field of research.
Author Response
We thank you for your clear comments on our manuscript. We are sorry to hear that, according to your review, our study suffered major flaws. However, we would like to furthermore justify our choice of the study design.
Cervical cancer screening detects precancerous lesions, before they progress to invasive cancer. Therefore, screening for cervical cancer is associated with a reduction in both, the incidence and mortality, of cervical cancer (1—3). This is possible due to the natural history of the disease: it develops over the years through a precancerous stage, preceded by a persistent infection with high‐risk HPV (4, 5). The precancerous lesion can be detected by screening and treated before it progresses to cancer.
The fact that cervical cancer can indeed be avoided through screening is an important notion and is now outlined in the beginning of the introduction (lines 33—37).
We believe that our results are a valid contribution to the literature. With a matched case-control -design and adjustment for appropriate variables, we were able to produce valid and up-to-date estimates for the protective effect of testing against cancer. In addition, our study considered the effectiveness of testing outside the screening program, a topic not well studied before. We have also highlighted this point a bit in the discussion (line 208).
We hope that these additions will justify our objective and study design, and that our paper could be considered for publication.
References
- IARC (2022). Cervical cancer screening. IARC Handb Cancer Prev. 18:1–456. Available from: https://publications.iarc.fr/604
- Peirson, L., Fitzpatrick-Lewis, D., Ciliska, D. et al. Screening for cervical cancer: a systematic review and meta-analysis. Syst Rev 2, 35 (2013). https://doi.org/10.1186/2046-4053-2-35
- Jansen, E.E.L.; Zielonke, N.; Gini, A.; Anttila, A.; Segnan, N.; Vokó, Z.; Ivanuš, U.; McKee, M.; de Koning, H.J.; de Kok, I.M.C.M.; et al. Effect of Organised Cervical Cancer Screening on Cervical Cancer Mortality in Europe: A Systematic Review. European Journal of Cancer 2020, 127, 207–223, doi:10.1016/j.ejca.2019.12.013.
- Ho, G. Y., Burk, R. D., Klein, S., Kadish, A. S., Chang, C. J., Palan, P., Basu, J., Tachezy, R., Lewis, R. and Romney, S. Persistent genital human papillomavirus infection as a risk factor for persistent cervical dysplasia. J Natl Cancer Inst 1995; 87(18): 1365‐1371.
- Koshiol J, Lindsay L, Pimenta JM, et al. Persistent human papillomavirus infection and cervical neoplasia: a systematic review and meta-analysis. Am J Epidemiol 2008;168:123–37.
Reviewer 2 Report
I thank the academic editor for giving me the pleasure of reviewing this interesting paper in which the authors conduct a case-control study on the effectiveness of the PAP-test and HPV-test in the Finnish population, over a period of about 20 years. on data obtained both from national screening programs and from data outside the screening programs. I believe, overall, that the paper is well structured and written. It is very understandable that from the results the authors found a greater ability of the PAP test to detect cervical squamous lesions rather than adenocarcinoma. Furthermore, it is very interesting to note that no significant improvements were found between 3 and 5 years screening.
I share, among other things, that a PAP test can be done once every 5 years compared to the canonical 3 years, with a negative HPV test in between. The authors' findings suggest this form of screening.
I would suggest the authors to implement the introduction section by obtaining a small paragraph on the most up-to-date knowledge regarding cervical cancer, HPV, histopathological classification of carcinoma, etc. as this could make, in my opinion, the paper even more wide-ranging. From a Medline search I would suggest the following papers to the authors:
Cervical Cancer Screening. J Midwifery Womens Health. 2022 Mar;67(2):285-286. doi: 10.1111/jmwh.13347. Epub 2022 Feb 20. PMID: 35187782.
Tabibi T, Barnes JM, Shah A, Osazuwa-Peters N, Johnson KJ, Brown DS. Human Papillomavirus Vaccination and Trends in Cervical Cancer Incidence and Mortality in the US. JAMA Pediatr. 2022 Mar 1;176(3):313-316. doi: 10.1001/jamapediatrics.2021.4807. PMID: 34842903; PMCID: PMC8630656.
Arezzo F, Cormio G, Loizzi V, Cazzato G, Cataldo V, Lombardi C, Ingravallo G, Cicinelli E. HPV-Negative Cervical Cancer: A Narrative Review. Diagnostics (Basel). 2021 May 26;11(6):952. doi: 10.3390/diagnostics11060952. PMID: 34073478; PMCID: PMC8229781.
Bewley S. HPV vaccination and cervical cancer screening. Lancet. 2022 May 21;399(10339):1939. doi: 10.1016/S0140-6736(22)00110-6. PMID: 35598621.
Cascardi E, Cazzato G, Daniele A, Silvestris E, Cormio G, Di Vagno G, Malvasi A, Loizzi V, Scacco S, Pinto V, Dellino M. Association between Cervical Microbiota and HPV: Could This Be the Key to Complete Cervical Cancer Eradication? Biology (Basel). 2022 Jul 26;11(8):1114. doi: 10.3390/biology11081114. PMID: 35892970; PMCID: PMC9351688.
Zaccaro R, Maggipinto K, Giacomino ME, De Nicolò M, De Summa S, Cazzato G, Scacco S, Malvasi A, Pinto V, Cicinelli E, Carriero C, Di Vagno G, Cormio G, Genco CA. Communications Is Time for Care: An Italian Monocentric Survey on Human Papillomavirus (HPV) Risk Information as Part of Cervical Cancer Screening. J Pers Med. 2022 Aug 26;12(9):1387. doi: 10.3390/jpm12091387. PMID: 36143172; PMCID: PMC9505682.
Finally, to further improve the quality of the paper, I suggest the authors to add explanatory images of a PAP-test, and, by way of example, a negative PAP-test, a doubtful PAP-test (ASC-US, ACG), and a frankly positive PAP test. After these improvements I believe that the paper will be accepted.
Author Response
We thank you for your encouraging comments and valuable suggestions, which will further improve our manuscript.
We have now added a paragraph on relevant information on HPV, the natural history of cervical cancer and its prevention (e.g. vaccination) in the introduction (lines 33—40). However, to keep the introduction clear-cut, we wrote in quite a general level.
Thank you for giving me the chance to read these interesting references. We used some of the papers in the introduction (Tabibi et al., Bewley et al.; line 38), when considering the impact of HPV-vaccination on cervical cancer prevention.
It would be a great addition to include explanatory images of how different Pap test results look like in reality. However, unfortunately we did not have access/permit on such images within the agreed timetable, that would be of sufficient quality. However, we added a reference (Cervical Cytology Bethesda System Atlas) in the methods section, when describing the screening result management policy in the Finnish program (lines 86—88).
I hope these additions further improved our paper so that it could be accepted for publication.
Reviewer 3 Report
The paper entitled “Effectiveness of cervical testing in and outside the screening program- a case-control study” examined the effect of testing in and outside the Finnish screening program on the risk of cervical cancer. The register-based study sample included 1,677 diagnosed cervical cancer cases and 16,738 controls. It was concluded that tests done in and/or outside the program were effective in lowering the risk of cervical cancer.
Overall, the article is very well written and organised, easily readable, and clear. The authors have collected extensive and comprehensive data using available registries for cervical cancer cases and population data for controls. The experimental design is appropriate and well planned, with large sample size. The methods and type of data acquired are well described. The statistical analyses are suitable. The results are presented clearly and in an organised manner. Although the effectiveness of Pap versus HPV test was not determined, the reason for this was discussed. The figures and tables are appropriate and self-explanatory. The discussion is good and relevant. The questions that I had were alluded to in the discussion. The authors also put forward the limitations of the study which are well written. As there is scarce literature on similar case-control studies, direct comparisons are difficult. Nevertheless, many aspects of the results are well discussed, and recommendations are valid and well thought out. Many of the references are self-citations, however, they are all relevant to the article.
I would like to suggest that the authors include a mention of the HPV vaccination program in Finland if any. Also, what would they recommend as the best and most cost-effective test for a national screening program in their country i.e. Pap or HPV test?
Line 158, the total percentage of SQCC (59%) and adenocarcinoma (31%) cases is 90%. What type of tumours was the rest of the 10%?
Minor spell checks eg in Figure 3, spelling of adenocarcinoma and SQCC invasive
Author Response
We thank you for your careful review and all the helpful comments, as well as your encouraging words on our manuscript.
We added a sentence in the discussion section (lines 289—290) on the vaccination program in Finland – an important background information. We also discussed the advantages and disadvantages of HPV testing as the primary method in the screening program (lines 277—280).
The remaining 10% of the tumors 9% were other or unspecified carcinomas of the cervix, (such as carcinoma NOS, malignant neoplasms and adenosquamous carcinomas), and 1% were sarcomas or other unspecified malignancies. We have now clarified this in the results section (lines 171—172).
Thank you for the remarks on the spelling errors, which have now been corrected in Figure 3.
We hope that after these corrections the paper would reach sufficient quality for acceptance.
Round 2
Reviewer 1 Report
I appreciated the authors’ response.
My questions and concerns are regarding association analysis. How many logistic regressions did the authors perform? What are the p-values? Did you consider multiple testing problems?
Author Response
We thank you for the important question.
We performed altogether 42 logistic regression models: 4 for the main analysis (for two test intervals and for two test categorizations), 3*4 = 12 for the morphology subtypes and 13*2 = 26 for the age groups.
We reported 95% confidence intervals instead of p-values, since they also represent random error: For a confidence interval that does not include one, the p-value would be <0.05. The confidence intervals can be seen in the tables and figures.
We did not correct for multiple testing in these analyses. Hence, there could be a risk of some of the results being significant by chance. On the other hand, our subgroup analyses were not independent and thus the conventional methods would have led to type II errors (1). The subgroups in our study were biologically plausible and carefully defined in advance, making this type of analysis appropriate (2). In addition, our conclusions relied on results that were particularly clear, consistent and in line with previous literature.
We have now mentioned in the methods section that no adjustment for multiple comparisons was performed in these analyses (lines 118—119). We hope that this mention is sufficient, and that the reader can further evaluate the possible impact on the results and conclusions.
References:
- Bland J M, Altman D G. Multiple significance tests: the Bonferroni method BMJ 1995; 310 :170
- Fayers PM, King MT. How to guarantee finding a statistically significant difference: the use and abuse of subgroup analyses. Qual Life Res. 2009 Jun;18(5):527-30.
Reviewer 2 Report
The Authors addressed my suggestions. Manuscript can be now accepted.
Author Response
We thank you for your valuable expert review on our manuscript.